# “Try to Build This Bunny as Fast as Possible without Using Red or Pink Bricks”: How Simple Assembly Tasks Might Aid in Detecting People with Mild Cognitive Impairment and Dementia

**DOI:** 10.3390/brainsci13121693

**Published:** 2023-12-08

**Authors:** Wolfgang Trapp, Andreas Heid, Susanne Röder, Franziska Wimmer, Helmar Weiß, Göran Hajak

**Affiliations:** 1Department of Psychiatry, Sozialstiftung Bamberg, St.-Getreu-Straße 18, 96049 Bamberg, Germany; andreas.heid@sozialstiftung-bamberg.de (A.H.); susanne.roeder@sozialstiftung-bamberg.de (S.R.); franziska.wimmer@sozialstiftung-bamberg.de (F.W.); helmar.weiss@sozialstiftung-bamberg.de (H.W.); goeran.hajak@sozialstiftung-bamberg.de (G.H.); 2Department of Physiological Psychology, Otto-Friedrich University Bamberg, Markusplatz 3, 96045 Bamberg, Germany

**Keywords:** dementia, mild cognitive impairment, Mini-Mental State Examination, Frontal Assessment Battery, executive function, apraxia, assembly test, Lego^®^ building blocks

## Abstract

(1) Background: Dementia and mild cognitive impairment (MCI) are still underdiagnosed in the general population. Impaired odor identification has been identified as an early marker of MCI and dementia. We aim to investigate whether short tasks, in which simple forms must be assembled from single building blocks based on a template or while considering specific re-strictions, could increase the diagnostic quality of established cognitive screening tests in detecting MCI or dementia. (2) Methods: A brief assembly test, where participants had to assemble simple animal shapes from Lego^®^ Duplo^®^ building blocks, the Frontal Assessment Battery, and the Mini-Mental State Exam (MMSE) were administered to a consecutive series of 197 patients (89 with mild dementia, 62 with mild cognitive impairment, and 46 without cognitive impairment) referred for neuropsychological testing. (3) Results: Both participants with dementia and with MCI performed badly in the assembly tasks. The assembly tasks and the Frontal Assessment Battery were substantially correlated. Complementing MMSE scores with the assembly tasks improved the diagnostic accuracy of individuals with dementia and MCI. (4) Conclusions: People with suspected dementia or MCI may already benefit from simple assembly tasks. Although these tests require little additional time, they can notably increase sensitivity for dementia or MCI.

## 1. Introduction

Dementia, according to the International Classification of Diseases (ICD-11) by the World Health Organization, is characterized by a substantial deterioration in an individual’s cognitive functioning across two or more domains unrelated to typical aging and significantly impeding daily activities [1]. At the moment, about 55 million people worldwide [2] are affected. The anticipated increase is projected to surpass 150 million individuals globally by 2050 [3].

However, an early diagnosis of dementia is crucial for several reasons. First, interventions could be initiated to slow the progression of cognitive deficits. Second, it would be possible to implement care plans while patients still have the legal capacity. Third, the onset of institutionalization could be delayed. Additionally, research indicates that these interventions improve the overall quality of life and postpone the need for institutional care [4,5,6]. Nevertheless, current numbers indicate that less than half of the people in the general population with dementia have received a formal diagnosis [7].

The generic concept of executive function (EF) usually covers a range of individual cognitive functions, such as switching between different subgoals, logical reasoning, activating content in working memory, and the ability to use these functions flexibly to achieve a desired goal [8]. The involved brain regions include the prefrontal cortex, the parietal cortex, the basal ganglia, the cerebellum, and the thalamus [9,10]. Executive impairment is frequent in frontotemporal dementia (particularly its behavioral variant) and in mild cognitive impairment [11], but also occurs, for example, in cerebrovascular dementia [12], Lewy body dementia [13], and also, although usually not as an early sign, in Alzheimer’s disease [14].

Apraxia, defined as the impaired ability to willingly perform planned sequences of actions consisting of skilled or learned movements not caused by elementary motor, sensory, or coordination deficits [15], is not only found in frontotemporal dementia but also occurs in Alzheimer’s disease during their early stages [16,17]. Very early on, apraxia was classified into ideomotor (e.g., imitation of gestures), ideational (inability to sequence a series of actions correctly), and limb-kinetic (impairments in precisely coordinated finger and hand movements) subtypes [18]. Especially in ideational apraxia, the underlying brain areas considerably overlap with the affected areas in executive dysfunction [19].

While neuropsychological test batteries to test for neurocognitive disorders typically include tests of executive functions, empirically validated measures of apraxia developed for people with suspected dementia are rare. Therefore, it could make sense to provide combined tasks that require praxis movements, on the one hand, and executive functions, on the other hand, as part of dementia diagnosis.

This paper investigates whether short assembly tasks, in which simple forms must be assembled from single building blocks based on a template or while considering specific restrictions, could increase the diagnostic quality of established cognitive screening tests.

In particular, we test the hypothesis that (1) individuals with dementia and MCI perform worse in these assembly tasks than healthy older adults; (2) there is evidence of construct validity (i.e., the assembly tasks show higher correlations with neurocognitive tests tapping executive function or apraxia); and (3) the assembly tasks improve diagnostic classification using the MMSE.

## 2. Materials and Methods

### 2.1. Participants

A consecutive sequence of patients, routinely referred for neuropsychological testing due to suspected cognitive decline or based on their request, was enlisted from the departments of geriatric internal medicine and geriatric psychiatry at a general hospital in Bamberg, Germany. All of them underwent routine laboratory screening, including thyroid function parameters, lues serology, B12, and folic acid levels, a cranial computer tomography (CT) or magnetic resonance imaging (MRI) scan, EEG, ECG, and a thorough neuropsychological, psychiatric, neurological, and physical examination to secure a proper diagnosis of MCI, mild dementia, or to exclude other causes of cognitive decline.

The decision of whether the examined patient had dementia or mild cognitive impairment (MCI) was reached through a multidisciplinary meeting, wherein the ICD-10 criteria for dementia diagnosis and additional established criteria [20,21,22,23,24,25,26] for diagnosing MCI and specific dementia subtypes were applied. Individuals diagnosed with moderate or severe dementia, as well as patients displaying notable depressive symptoms, were excluded from participation.

In total, a convenience sample of 197 participants was recruited: 62 (35 females) with MCI, 89 (50 females) with mild dementia (DEM; 50 with Alzheimer’s disease, and 39 with other types of dementia), and 46 without cognitive impairment (24 females). The latter group was, therefore, included as a clinical control sample (CNT). All recruited patients consented to participate in the study. Participants were not allocated randomly to the three groups but based on their diagnosis.

### 2.2. Neurocognitive Tests, Assembly Task, and Symptom Measures

The Mini-Mental State Examination (MMSE [27]) was conducted as part of an extensive neurological test battery. This battery included the German version of the Consortium to Establish a Rationale in Alzheimer’s Disease diagnostic neuropsychological battery (CERAD-Plus [28]), the Bamberg Dementia Screening Test (BDST [29]), and the German version of the Frontal Assessment Battery (FAB-D [30]). All participants underwent these assessments.

For the assembly test, a selection of Lego^®^ Duplo^®^ building blocks was first laid out on a table in front of the participants, as shown in Figure 1a. Then, the first figure (dog, see Figure 1b) was placed in front of the participants, and they were asked to assemble the exact figure from the bricks as quickly as possible (assembly task 1). The experimenter then disassembled the figure again, and the second figure (bunny, see Figure 1c) was presented with the request to assemble it as quickly as possible (assembly task 2). Again, the figure was disassembled by the experimenter, the first figure (dog, see Figure 1b) was presented again, and participants were asked to assemble a figure with the same shape without using white or black bricks (assembly task 3, see Figure 1d for the correct solution). The figure was disassembled one last time, and the second figure (bunny, see Figure 1c) was presented again, this time with the instruction not to use red or pink bricks (assembly task 4, see Figure 1e for the correct solution). The time required for assembly was recorded for each of the four subtasks described above. If the figures assembled by the participants did not correspond to the template, the participants were informed and asked to correct their figures accordingly.

In addition, all patients completed the German short version of the Geriatric Depression Scale (GDS) [31], a brief screening tool for assessing depressive symptoms in older adults. Participants with GDS scores exceeding 5, indicative of potential depression, were excluded.

### 2.3. Statistical Analyses

Univariate analyses of variance to compare age, GDS scores, and years of education in the three diagnostic groups (CNT, MCI, and mild dementia) were performed. Likewise, univariate analyses of variance using Scheffé a posteriori comparisons were conducted for the MMSE, FAB-D, and the four tasks of the assembly test.

Pearson’s correlation coefficients were calculated between the four tasks of the assembly test and the MMSE, FAB-D, and specific CERAD-Plus subtests, including verbal learning, verbal recall, verbal recognition, Trail Making Test A, and Trail Making Test B to assess the concurrent and discriminant validity of the assembly test.

In order to compare the diagnostic performance of the MMSE, the FAB-D, and the four tasks of the assembly test, five stepwise logistic regression analyses were conducted using the diagnostic groups:CNT vs. mild DEM;CNT vs. mild Alzheimer’s disease;CNT vs. mild dementia (no Alzheimer’s disease);CNT vs. MCI;CNT vs. MCI or mild dementia as the dependent variables.

Finally, to obtain a first impression regarding the diagnostic quality of the assembly test, MMSE scores and the completion time in seconds for assembly task 4, which was selected in the equation during the stepwise linear regression analyses, were used to plot receiver operating characteristic (ROC) curves of sensitivity against 1-specificity for CNT vs. MCI or mild dementia subjects. The optimum cutoff scores for the MMSE and assembly task 4 were determined using the Youden index, and sensitivity and specificity were computed separately for each of the two tests based on the cutoff scores found and again for the case of at least one positive result in one of the two measures.

## 3. Results

### 3.1. Sample Characteristics

No significant differences were found in the three groups (CNT, MCI, and DEM) concerning age, GDS scores, and years of education (see Table 1 for more detailed information about the sample).

However, the three groups differed significantly in their MMSE, FAB-D, and assembly task scores. Scheffé a posteriori comparisons were significant for the comparison between CNT and DEM and between MCI and DEM (*p* < 0.0005 each) for all measures but not for the comparison between CNT and MCI, except for the scores for the FAB-D and assembly task 4 (“build the bunny without using red or pink bricks”) of the assembly test (*p* = 0.023 and 0.002).

### 3.2. Validity of the Assembly Tasks

As it can be seen from Table 2, all assembly tasks correlate higher with the FAB-D than with the MMSE. In addition, higher correlations of assembly tasks 2 and 4 (assembly with constraints) with the TMT B and higher correlations of assembly tasks 1 and 3 (assembly without constraints) with the TMT A were obtained. Furthermore, the correlations with the CERAD verbal memory tasks were lower than those with the TMT tests.

### 3.3. Diagnostic Performance of the MMST, the FAB-D, and the Assembly Tasks

Table 3 shows the results of the stepwise regression analyses for the different diagnostic scenarios.

While the MMST is part of the prediction equations for dementia or Alzheimer’s disease, assembly task 4 of the assembly test is part of the equation for predicting the presence of non-Alzheimer’s dementia or MCI. The FAB is found in all prediction equations, and for the clinically relevant case of the prediction of any cognitive impairment (DEM or MCI), all three tests make a significant predictive contribution.

ROC analyses determined optimal cutoff scores of ≥90 s for assembly task 4 of the assembly test (bunny2, sensitivity = 62.3% and specificity = 87.0%) and ≤27 for the MMSE (sensitivity = 60.3% and specificity = 82.6%). When both measures are combined (suspected Alzheimer’s disease or MCI when either MMSE or assembly task 4 is positive), sensitivity increases to 84.1% while specificity decreases to 73.0%.

## 4. Discussion

This paper aimed to clarify the usefulness of short assembly tasks that require executive functions and ideomotor praxis movements for the early detection of cognitive decline in dementia.

A simple block-building assembly task, like the one presented in this article, is simple, quick to administer, and engages a broad spectrum of cognitive functions, making it a valuable addition to the existing neurocognitive screenings for cognitive decline.

First, spatial awareness might be engaged, as individuals must visualize how the pieces fit together in three-dimensional space. Second, handling and connecting the blocks involves precise sequences of movements and hand–eye coordination. Both abilities might be impaired in persons with beginning apraxia that may occur in the early stages of dementia [16,17].

Also, working memory skills might be involved, as participants need to recall the precise instructions about the arrangement of the blocks and the ancillary conditions, like not using bricks of a particular color in assembly tasks 3 and 4.

Finally, executive functions are needed to complete assembly tasks. The three-component model [32] that has been very influential differentiates between updating (adding new relevant information to memory or removing information that is no longer relevant from working memory), shifting (i.e., switching between mental sets), and the inhibition of prepotent automatic responses as critical aspects of executive function. The block-building assembly task covers all three aspects. For example, depending on the current progress, the participants must add new content (e.g., specific characteristics of the next building block needed) to their working memory storage buffers and remove already used building blocks from their working memory storage buffers.

As stated in the Section 1, broader concepts of EF include cognitive functions, like switching between different subgoals, logical reasoning, and the ability to use these functions flexibly [8].

Especially in assembly tasks 3 and 4, the participants must pay attention to the outlines of the building blocks and the target figure and abstract from the color and design of the building blocks, as the finished figure no longer looks like a dog or a bunny.

Therefore, it is no surprise that there was a significant difference in the processing times between people with and without cognitive impairments. People with cognitive impairments took longer to complete the assembly tasks.

Also, there were clear indications of validity and construct validity. Correlations with neuropsychological tests designed to capture similar cognitive domains were more pronounced than correlations with other neurocognitive tests. For example, the assembly tasks of the assembly test correlated higher with the FAB than with the MMSE, and higher correlations were found with the two Trail Making tests than with verbal memory tasks, whereby the highest correlation resulted between assembly tasks 3 and 4, which place the most significant demand on executive functions, and the FAB. In addition, both assembly tasks correlate higher with TMT B than TMT A, whereas assembly tasks 1 and 2, which are less demanding on EF and therefore focus more on visual–spatial processing and processing speed, correlate more strongly with TMT A.

In addition, the results of the logistic regression analyses indicate that the prognosis of whether there is a decline in cognitive performance can be improved by assembly task 4, especially when people with mild dementia but not with Alzheimer’s disease and/or people with MCI are considered. Notably, the MMSE does not appear in the equations when other types of dementia than Alzheimer’s disease or MCI are predicted. These results could be due to the low sensitivity of the MMSE concerning the detection rate of people with MCI. Maybe in non-Alzheimer’s disease and MCI, the semantic and episodic memory items and the orientation questions of the MMSE, which do not overlap with the tasks from the FAB or the block-building assembly tasks, are not challenging enough to detect mildly cognitively impaired persons. Consequently, the high ceiling effects of MMSE in MCI patients compared to MoCA have been reported [33]. Assembly task 4 places the most significant demands on executive functions. The bunny shape that must be assembled is more complex than the dog shape of assembly task 3 as six instead of four building blocks must be assembled while participants must inhibit prepotent responses (choosing the original building blocks of the bunny).

The cognitive decline detection rate could thus improve in clinical practice even by adding a very brief assembly task. For the dataset analyzed in this paper, an increase in sensitivity of almost 25% was found when, in addition to the MMST, assembly task 4, which takes less than 2 min to complete, is administered and the optimal cut-off score of ≤27 for the MMSE is used.

Trivially, using an additional test should increase sensitivity at the expense of reducing specificity (see [34] for a more detailed discussion) because the additional test should detect some subjects not correctly identified by the first test. Conversely, there is a heightened risk of erroneously categorizing unimpaired individuals as impaired in either of the two methods. However, the sensitivity increase in our case exceeds the specificity decrease when the MMSE and assembly task 4 are combined.

A critical point to consider is that limitations in motor skills and speed can also decline with age for other reasons. Nevertheless, a high specificity was found for assembly task 4 of the assembly test. This may be partly due to the large size of the building blocks (the average size of the bricks was 72 cm^3^/4.4 in^3^). Therefore, the additional cognitive demands on executive function and praxis movement (recognizing the bricks that make up the pattern, picking identical bricks or bricks of the same shape, inhibiting the tendency to use identical bricks in assembly tasks 3 and 4, and correctly performing a series of successive motor actions) might have been the main reason for the increase in processing time in people with cognitive impairment.

To the best of our knowledge, no block-building assembly tasks like the one introduced in this paper have been used to diagnose EF dysfunction in people with dementia or other mental disorders. Most authors investigated block-building assembly tasks in children where they could show that executive functioning and mental rotation capabilities were linked with performance in these assembly tasks [35,36]. Some authors even hypothesize that block-building assembly tasks might be utilized to promote or measure Theory of Mind and emotion understanding [37,38].

A crucial limitation of the presented study is the comparison of assembly tasks with the MMSE, which is deemed unsatisfactory, particularly in distinguishing between mild cognitive impairment (MCI) and cognitively non-impaired individuals [39]. Therefore, future studies should include data from more sensitive cognitive screening tests, like the Montreal Cognitive Assessment (MoCA, [40]), the Addenbrooke’s Cognitive Examination III (ACE-III, [41]), or the Test Your Memory (TYM [42]).

The data presented in this paper were derived from a clinical sample of patients referred for neuropsychological testing. While this setting may be appropriate in some instances (such as in a geriatric ward of a general hospital where a rapid assessment could be advantageous), it resulted in a high proportion of participants with the target conditions (dementia or MCI). Given our small sample size, the results need to be cross-validated, preferably in a population-based sample.

## 5. Conclusions

This study shed light on the potential of a simple block-building assembly task as a valuable tool for the early detection of cognitive decline in dementia. The task engages a spectrum of cognitive functions, including spatial awareness, hand–eye coordination, working memory, and executive functions. The observed differences in processing times between individuals with and without cognitive impairments underscore the task’s sensitivity to cognitive decline.

The findings demonstrate clear indications of validity and construct validity, with stronger correlations observed with neuropsychological tests capturing similar cognitive domains.

The study suggests that incorporating a brief assembly task alongside traditional cognitive screenings, such as the MMSE, could enhance sensitivity in detecting cognitive impairment, especially in individuals with mild dementia and those with mild cognitive impairment (MCI). While acknowledging the potential trade-off between sensitivity and specificity, the increase in sensitivity appears promising.

However, the study acknowledges its limitations, including the reliance on a clinical sample and the need for cross-validation in a population-based sample. Future research should explore the integration of more sensitive cognitive screening tests and further validate the utility of block-building assembly tasks in diverse populations. Despite these limitations, the study opens a new avenue for exploring the role of such tasks in diagnosing the dysfunction of executive functions in individuals with dementia and other mental disorders, offering a novel perspective for early intervention and assessment.

## Figures and Tables

**Figure 1 brainsci-13-01693-f001:**
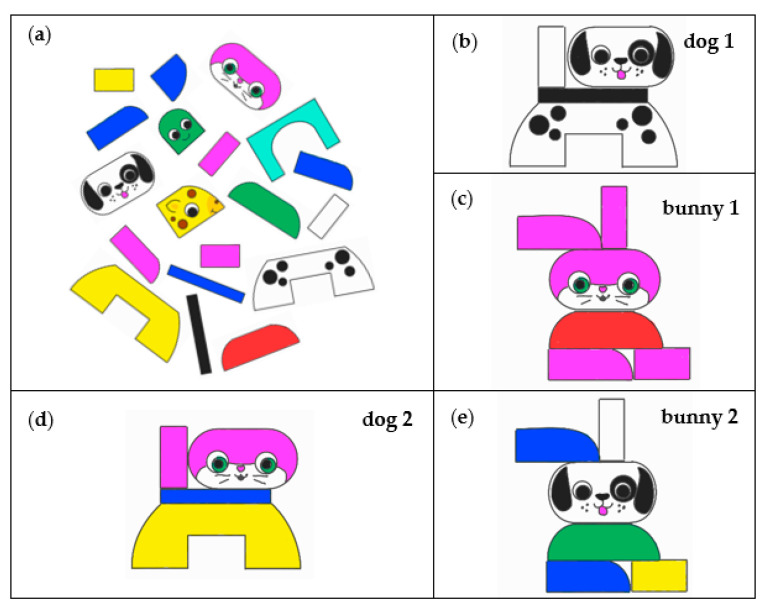
Assembly test. (**a**) Building blocks provided to the participants, (**b**,**c**) templates for the assembly tasks 1 and 2, and (**d**,**e**) solutions for the assembly tasks 3 and 4.

**Table 1 brainsci-13-01693-t001:** Sample characteristics. SD: standard deviation; CNT: clinical control sample; DEM: mild dementia; MCI: mild cognitive impairment; GDS: Geriatric Depression Scale; MMSE: Mini-Mental Status Examination; Dog 1–Bunny 2: assembly task 1 to 4 of the assembly test.

	CNT (n = 46)	MCI (n = 62)	DEM (n = 89)	Analysis of Variance
	mean (SD)	mean (SD)	mean (SD)	F_(2,194)_ (*p*)
Age	69.65 (10.03)	70.55 (9.58)	69.06 (8.72)	0.470 (0.626)
Years of education	13.48 (1.97)	13.79 (2.12)	13.18 (1.75)	1.793 (0.169)
GDS	4.28 (3.42)	4.52 (3.52)	4.48 (3.98)	0.060 (0.942)
MMSE	28.39 (1.20)	28.03 (1.34)	25.16 (2.62)	56.003 (<0.0005)
FAB-D	17.11 (1.16)	15.76 (2.06)	12.56 (3.18)	59.513 (<0.0005)
Dog 1	18.41 (10.74)	26.60 (15.29)	40.70 (25.15)	21.767 (<0.0005)
Dog 2	25.48 (20.28)	40.39 (30.65)	70.99 (55.43)	20.314 (<0.0005)
Bunny 1	32.67 (18.27)	42.10 (24.37)	60.60 (34.30)	16.694 (<0.0005)
Bunny 2	56.67 (28.14)	88.29 (43.17)	125.43 (51.93)	37.968 (<0.0005)

**Table 2 brainsci-13-01693-t002:** Correlations of the assembly tasks with cognitive measures. R: Pearson’s correlation coefficient; MMSE: Mini-Mental Status Examination; Dog1–Bunny2: assembly task 1 to 4 of the assembly test; CERAD-Plus: German version of the Consortium to Establish a Rationale in Alzheimer’s Disease diagnostic neuropsychological battery; FAB-D: German version of the Frontal Assessment Battery; TMT: Trail Making Test.

		Assembly Test			
		Dog 1	Dog 2	Bunny 1	Bunny 2
		r (*p*)	r (*p*)	r (*p*)	r (*p*)
MMSE		−0.36 (<0.0005)	−0.46 (<0.0005)	−0.33 (<0.0005)	−0.36 (<0.0005)
CERAD-Plus	Verbal learning	−0.27 (<0.0005)	−0.34 (<0.0005)	−0.26 (<0.0005)	−0.34 (<0.0005)
	Verbal recall	−0.27 (<0.0005)	−0.30 (<0.0005)	−0.24 (<0.0005)	−0.33 (<0.0005)
	Verbal recognition	−0.18 (0.011)	−0.25 (0.001)	−0.15 (0.033)	−0.18 (0.011)
	TMT A	−0.54 (<0.0005)	−0.47 (<0.0005)	−0.58 (<0.0005)	−0.55 (<0.0005)
	TMT B	−0.51 (<0.0005)	−0.52 (<0.0005)	−0.46 (<0.0005)	−0.63 (<0.0005)
FAB-D		−0.45 (<0.0005)	−0.52 (<0.0005)	−0.42 (<0.0005)	−0.56 (<0.0005)

**Table 3 brainsci-13-01693-t003:** Results of the logistic regression analyses to predict cognitive impairment. MMSE: Mini-Mental Status Examination; AT 4: assembly task 4 of the assembly test; OR: odds ratio; SE: standard error; β: standardized regression coefficient.

Dependent Variable	Variables in the Equation	β	SE (β)	*p*	OR	−2 Log- Likelihood	Nagelkerkes’s R^2^
CNT vs. DEM	MMSE	−1.004	0.245	<0.0005	2.732	58.809	0.791
	FAB-D	−1.075	0.237	<0.0005	2.933		
CNT vs. DEM (Alzheimer’s disease)	MMSE	−1.253	0.321	<0.0005	3.497	38.343	0.836
	FAB-D	−1.153	0.365	0.002	3.165		
CNT vs. DEM (other types of dementia)	AT 4 (bunny2)	0.038	0.013	0.003	1.039	29.768	0.859
	FAB-D	−1.195	0.329	<0.0005	3.300		
CNT vs. MCI	AT 4 (bunny2)	0.022	0.002	<0.0005	1.022	118.859	0.311
	FAB-D	−0.422	0.004	0.047	1.524		
CNT vs. (MCI or DEM)	MMSE	−0.465	0.150	0.002	1.592	130.803	0.520
	AT 4 (bunny2)	0.021	0.006	0.001	1.022		
	FAB-D	−0.545	0.142	<0.0005	1.724		

## Data Availability

The dataset used and analyzed in the current study is available from the corresponding authors upon reasonable request. The data are not publicly available due to privacy reasons.

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
