# Peer review of "“Try to Build This Bunny as Fast as Possible without Using Red or Pink Bricks”: How Simple Assembly Tasks Might Aid in Detecting People with Mild Cognitive Impairment and Dementia"

_brainsci, 2023, doi:10.3390/brainsci13121693_

Round 1
Reviewer 1 Report
Comments and Suggestions for Authors
Dear Authors, some parts of the article need clarification:
1) The sentences "A senior psychiatrist saw all patients. The decision as to whether the examined patient ......: " do not explain who made the decision about including patients in the study. We only know that the psychiatrist saw a patient
2) Why did you decide to use MMSE instead of MOCA? In my opinion, MOCA would be more appropriate to asses wider spectrum of cognitive functions
3) in my opinion, the recruitment process for the study is not sufficiently described. How were patients assigned to the different groups? Randomly ? This detailed description should be provided in the Participants section. I would also recommend moving information from lines 146-149 to the Participants section
4) What relevance to the purpose of the work is the information given in lines 80-81: . All of them underwent routine laboratory screening, including thyroid function parameters, lues serology, B12, and folate acid levels..."?
5) did you have any drop-out during the study? How many participants were not included in the study due to some reasons? It would be great to see a flowchart showing that
6) I believe that introducing obvious information into a scientific article is unnecessary; please see lines 207-209: There was a clear difference in the processing times between people with and with- 207 out cognitive impairments. People with cognitive impairments took longer to complete 208 the assembly tasks.
Author Response
Dear reviewer,
thank you very much for your thorough review, and your many helpful comments. We think the quality of our manuscript has improved considerably after considering your hints.
1) The sentences "A senior psychiatrist saw all patients. The decision as to whether the examined patient ......: " do not explain who made the decision about including patients in the study. We only know that the psychiatrist saw a patient.
This is true. Thank you very much for this hint. As the participants were derived from a convenience sample (see 3)), all of the patients who consented were included in this study. But you're right. The sentence alone doesn't make much sense. That's why we deleted it.
2) Why did you decide to use MMSE instead of MOCA? In my opinion, MOCA would be more appropriate to asses wider spectrum of cognitive functions.
You are absolutely right. However, we did not administer the MOCA because we already used an extensive test battery, including the CERAD-Plus (which includes the MMSE), the FAB, and the BDST. We added that comparing the assembly task with the MOCA would have been valuable and have addressed this issue in the discussion section.
3) in my opinion, the recruitment process for the study is not sufficiently described. How were patients assigned to the different groups? Randomly ? This detailed description should be provided in the Participants section. I would also recommend moving information from lines 146-149 to the Participants section.
As stated in the “participants” section (lines 77-79), the participants were derived from a convenience sample of patients who were referred for neuropsychological testing on a routine basis because of a suspected cognitive decline or due to their wish. Therefore, no randomization was carried out, and the allocation to the individual groups was based on the diagnosed degree of cognitive impairment (mild dementia, MCI, or no cognitive impairment). We have added additional information about the type of the sample and about how the different groups were obtained. Furthermore, as you recommended, we have moved the information from lines 146-149 to the “participants” section.
4) What relevance to the purpose of the work is the information given in lines 80-81: . All of them underwent routine laboratory screening, including thyroid function parameters, lues serology, B12, and folate acid levels..."?
We did this to secure a proper diagnosis of MCI and mild dementia or to exclude other causes of cognitive decline. Thank you for this valuable hint. We have added this information.
5) did you have any drop-out during the study? How many participants were not included in the study due to some reasons? It would be great to see a flowchart showing that
Thank you very much. All the patients in our convenience sample consented to participate in the study. We have added this information to the manuscript. As no randomization or repeated testing was performed, we refrained from inserting a flow chart, as this would not provide any additional information. We hope this is okay for you.
6) I believe that introducing obvious information into a scientific article is unnecessary; please see lines 207-209: There was a clear difference in the processing times between people with and with- 207 out cognitive impairments. People with cognitive impairments took longer to complete 208 the assembly tasks.
Thank you very much. I think we have not expressed ourselves very skillfully at this point. We actually wanted to say that the apparent differences are statistically significant. We have changed the relevant sentence accordingly.
Reviewer 2 Report
Comments and Suggestions for Authors
This study found that administering a hands-on block-building task and interpreting its performance alongside the results of a traditional screening assessment was more helpful in screening for mild cognitive impairment and dementia. Block building is simple and quick to administer, and appears to have high ecological validity in that it requires direct action, suggesting that it can be administered alongside existing screening assessments. However, the current discussion needs to go into more detail about how the specific block-building tasks used in this study relate to cognitive function. The current discussion simply repeats the findings that there was a correlation.
Author Response
Dear reviewer,
thank you very much for your thorough review, encouraging words, and your helpful comments. We think the quality of our manuscript has improved considerably after considering your hints.
This study found that administering a hands-on block-building task and interpreting its performance alongside the results of a traditional screening assessment was more helpful in screening for mild cognitive impairment and dementia. Block building is simple and quick to administer, and appears to have high ecological validity in that it requires direct action, suggesting that it can be administered alongside existing screening assessments. However, the current discussion needs to go into more detail about how the specific block-building tasks used in this study relate to cognitive function. The current discussion simply repeats the findings that there was a correlation.
You are right. Thank you very much for this hint. We have extended the discussion section and added additional lines of information about the cognitive functions that might be engaged.
Round 2
Reviewer 1 Report
Comments and Suggestions for Authors
Dear Authors, thank you for al the improvements